# Progress on Diamane and Diamanoid Thin Film Pressureless Synthesis

**Fabrice Piazza** [1,*] , **Marc Monthioux** [2] , **Pascal Puech** [2], **Iann C. Gerber** [3] and **Kathleen Gough** [4]

1   Laboratorio de Nanociencia, Pontificia Universidad Católica Madre y Maestra, Santiago, Apartado Postal 822, Dominican Republic

2   Centre d'Elaboration des Matériaux et d'Etudes Structurales (CEMES), CNRS, Université de Toulouse, BP 94347, Toulouse CEDEX 4, 31055 Toulouse, France; marc.monthioux@cemes.fr (M.M.); pascal.puech@cemes.fr (P.P.)

3   Laboratoire de Physico-Chimie des Nano-Objets (LPCNO), CNRS, INSA, Université de Toulouse, 31400 Toulouse, France; igerber@insa-toulouse.fr

4   Department of Chemistry, University of Manitoba, Winnipeg, MB R3T 2N2, Canada; Kathleen.Gough@umanitoba.ca

*   Correspondence: fpiazza@pucmm.edu.do

**Abstract:** Nanometer-thick and crystalline $sp^3$-bonded carbon sheets are promising new wide band-gap semiconducting materials for electronics, photonics, and medical devices. Diamane was prepared from the exposure of bi-layer graphene to hydrogen radicals produced by the hot-filament process at low pressure and temperature. A sharp $sp^3$-bonded carbon stretching mode was observed in ultraviolet Raman spectra at around 1344–1367 $cm^{-1}$ while no $sp^2$-bonded carbon peak was simultaneously detected. By replacing bi-layer graphene with few-layer graphene, diamanoid/graphene hybrids were formed from the partial conversion of few-layer graphene, due to the prevalent Bernal stacking sequence. Raman spectroscopy, electron diffraction, and Density Functional Theory calculations show that partial conversion generates twisted bi-layer graphene located at the interface between the upper diamanoid domain and the non-converted graphenic domain underneath. Carbon-hydrogen bonding in the basal plane of hydrogenated few-layer graphene, where carbon is bonded to a single hydrogen over an area of 150 $\mu m^2$, was directly evidenced by Fourier transform infrared microscopy and the actual full hydrogenation of diamane was supported by first-principle calculations. Those results open the door to large-scale production of diamane, diamanoids, and diamanoid/graphene hybrids.

**Keywords:** diamane; diamanoid; bi-layer graphene; twisted bi-layer graphene; hot-filament; hydrogenation; UV Raman spectroscopy

## 1. Introduction

Genuine diamane consists of two crystalline $sp^3$–bonded carbon layers for which half of the carbon atoms are hydrogenated while the other half bond the two layers to each other [1,2]. The material stability was first predicted by Chernozatonskii et al. in 2009 [1]. According to calculations, diamane is a semiconducting material, with a direct wide band-gap, which is very attractive for nanoelectronics, band-gap engineering, and active laser medium in nanooptics [3,4]. Because of its expected high thermal conductivity, diamane may be used in thermal management devices [5]. Owing to the lower effective Bohr radius of defects and higher radiative electronic transition rate, calculations have shown that diamane is a better host system for single photon emission than diamond [6]. Diamane is also expected to be very strong; therefore, it may be very attractive for ultrathin protective coatings, ultrahigh-strength components in composite materials for aerospace applications for instance, and nano-electromechanical systems [5]. Because of the expected low friction coefficient of the hydrogenated surface, diamane may also be used to improve

the wear resistance of coated mechanical parts [2]. The expected strength, low coefficient of friction and biocompatibility could render diamane very competitive as a building material to make lower power and miniaturized electronics and biomedical devices [2]. A diamane nanoribbon resonator was calculated to possess a high natural frequency, high quality factor, high figure of merit, high in-plane stiffness, and, as opposed to graphene, to be free from the influence of the edge configuration; consequently, it would be better than single-layer graphene (1LG), bi-layer graphene (2LG), $MoS_2$, or other two-dimensional (2D) material resonators (at least for the temperature range from 1 K to 300 K) [7]. Hetero-structures of graphene and diamane would be attractive for tunnel devices, optical linear waveguides, high efficiency optoelectronic sensors, lithium batteries, and supercapacitors [3,5].

According to calculations, diamane could be produced by the chemisorption of hydrogen atoms on the "top" and "bottom" surfaces of 2LG and the subsequent interlayer bonding between $sp^3$–bonded carbon atoms ($sp^3$–C), for instance in hydrogen cold plasma [1,8,9]. Another possibility would be the conversion of 2LG on a metal surface into a bi-layer of $sp^3$-bonded carbon through hydrogenation of the top layer. In this case, strong hybridization between the $sp^3$ dangling bond orbitals and the metallic surface orbitals would stabilize the $sp^3$–bonded carbon layers [10]. In such a case, the material would differ from genuine diamane, as only one surface would be hydrogenated. However, until recently, there was no report of any experimental evidence of such a scenario. Until 2019, no experimental studies on the hydrogenation of 1LG nor 2LG had convincingly demonstrated diamane formation for instance through the detection of diamane features in Raman spectra, as reviewed in [2]. A structure transformation of few-layer graphene (FLG) into a diamond-like layer was reported in [11]; however, the crystal structure was not confirmed. Neither Raman spectroscopy nor electron diffraction results were reported. The material obtained in [11] may have been $sp^3$–C-rich amorphous carbon [12,13]. Temporary interlayer bonding between graphene layers was reported [14–16] but this was shown to happen only when graphene layers were exposed to very high pressures.

Recently, the synthesis of stable nanometer-thick and crystalline $sp^3$–bonded carbon was unambiguously shown for the first time [2,17]. This breakthrough was achieved via chemisorption of H radicals produced by the hot-filament process at low temperature and pressure on the basal planes of FLG and the subsequent interlayer bonding between ($sp^3$–C). The $sp^3$–C stretching peak was detected in visible and UV Raman spectra, but these reports failed to substantiate the formation of genuine diamane. The starting graphene material was poorly suitable, with too many layers ordered following the Bernal or the rotationally faulted (turbostratic) stacking sequences; the Raman spectroscopy lacked sufficient lateral resolution for precise confirmation [17,18]. Instead, surface-hydrogenated hybrid materials composed of crystalline $sp^3$–C sheet, called diamanoid [2], and graphenes were shown [17]. Following that work, the synthesis of fluorinated single-layer diamond (F–diamane), where carbon atoms are fluorinated rather than hydrogenated as in genuine diamane, was claimed from the fluorination, at low pressure, of AB-stacked 2LG [19]. However, the analytical method, based on the intensity of the electron diffraction spots was discussed to be poorly reliable for ultrathin films [17], and most importantly, no $sp^3$–C-related peak was observed in the Raman spectra [19]. Very recently, it was shown that the hot-filament-promoted hydrogenation process can be successfully used to produce genuine diamane from 2LG graphene, supposedly from regions where graphene sheets are AB- or AA-stacked [18]. Therein, UV Raman spectroscopy revealed the $sp^3$–C stretching mode peak in diamane over areas as large as several tens of square micrometers [18].

In this work, we have comprehensively reviewed the hydrogenation of 2LG by the chemisorption of H generated from the dissociation of $H_2$ in a hot filament reactor at low temperature and low pressure, and the subsequent structure conversion into genuine diamane. We also reviewed the formation of diamanoid/graphene hybrids from the replacement of 2LG with FLG. UV Raman spectroscopy, which was shown to be critical for the characterization of such $sp^3$–C nanomaterials [2,17,18], was used to track the structure conversion and the extension of the converted domains. Very low voltage (5 kV)

transmission electron microscopy (TEM) analysis was carried out, mostly in electron diffraction mode. FTIR microscopy was employed to track the formation of C–H bonds. First-principle calculations and simulated electron diffraction were used to explain our findings.

## 2. Materials and Methods

### 2.1. Pristine Graphene Materials

For the synthesis of diamane, as-received suspended 2LG graphene films deposited on 3 mm diameter gold Quantifoil TEM grids from Graphenea were used as starting graphene materials. Coverage of the grid was ~98%. The 2LG films were actually obtained from the successive transfer of two multi-domain 1LG individually grown by chemical vapor deposition (CVD), hence the resulting graphene stacking sequence varies randomly all over the film. However, considering the dimension of the 1LG layers (~3 mm in diameter) and their polycrystalline nature (20-µm-large grains for the most), chances for locally getting the favorable AB or AA stacking configurations [1,2] at micrometer scale are high [18].

For the synthesis of diamanoid/graphene hybrids, as received FLG films deposited on 3 mm copper TEM grids (2000 Mesh) from Graphene Supermarket (SKU # SKU-TEM-CU-2000-025) were used as graphene materials. The FLG films were grown by CVD from $CH_4$ at 1000 °C on Ni substrate and transferred onto a commercial TEM grid using a polymer-free transfer method to minimize contamination as described in Ref. [20]. FLG thickness is typically between 0.3–2 nm (1–6 monolayers); however, thicker films, up to 20 layers (or 10, depending on the folding configuration), were observed in TEM. Typical FLG TEM grid coverage is between 60 and 90%. The film is therefore polycrystalline in the in-plane direction, but the layers within each grain are likely to be Bernal (ABAB) stacked or rotationally faulted.

### 2.2. Hydrogenation Process

A commercial hot filament reactor was used for the hydrogenation process [21]. The specific experimental details related to the synthesis of diamane and diamanoid/graphene hybrids were previously disclosed [2,17,18]. Briefly, two tungsten wires were used as heating elements. Prior to mounting, the filaments were cleaned with acetone to avoid any carbon contamination. For the same reason, before the hydrogenation process, the filaments were not exposed to hydrocarbon gas for carburization or conditioning as is otherwise necessary for the conventional growth of diamond by hot filament CVD. The grids were placed vertically on the movable and fluid-cooled stainless steel substrate holder. The substrate holder and grid temperatures were estimated by a thermocouple tip located on top of the surface of the substrate holder located right under the filaments. After evacuating the chamber down to at least $10^{-1}$ Torr, a regulated flow of ultra-high purity $H_2$ gas was introduced from the top of the chamber. The pressure was regulated via an automatic valve located below the substrate holder. Once $H_2$ was introduced into the chamber and steady-state conditions were reached, the filament temperature was raised and maintained within the range of 2350–2590 °C. The pressure and flow were 50 Torr and 1 sccm, respectively. The distance between the substrate holder and the filaments was 22.8 mm. The resulting maximum substrate holder temperature was ~325 °C in the case of FLG, and 400 °C in the case of 2LG. Those estimations take into account a measured 50 °C substrate holder temperature variation across the grid [2]. The duration of the hydrogenation process was 6 min and 20 s.

### 2.3. Material Characterization

Raman spectroscopy is a critical technique to characterize carbon materials. Raman scattering from $sp^2$–C is a resonant process for visible wavelength excitation and would result in a 50 to 230-fold signal enhancement relative to that of non-resonant $sp^3$–C Raman scattering [22]; thus, it is necessary to employ UV excitation to obtain a more evenly weighted probing of $sp^3$–C and $sp^2$–C sites. By calculating the band-gap values of genuine

diamane and diamanoid with ABC stacking (both surfaces hydrogenated) from $G_0W_0$ calculations using the Heyd–Scuseria–Ernzerhof exchange-correlation functional, we have confirmed the necessity of using a deep UV excitation for Raman spectroscopy study of ultrathin and crystalline $sp^3$–C film [2]. The present Raman scattering measurements were performed in the backscattering configuration using the 244 nm line of an Ar ion laser (85 Lexel Second Harmonic Generation laser) as the excitation source. The backscattered light was analyzed using a ×40 objective micro-Raman spectrometer system (inVia Reflex, Renishaw) equipped with a stigmatic single-pass spectrograph including a holographic grating of 3600 grooves.mm$^{-1}$ and with a 1″ Peltier-cooled UV coated Deep Depletion CCD array detector. The entrance slit was set to 50 μm for a spectral resolution of 3.5 cm$^{-1}$. The laser power on the sample and the acquisition time were adjusted to obtain optimum signal without any sample alteration. Typically, laser power was ~1 mW, and exposure time to the laser was ~1 s. No visible damage and no change of the spectral profile were observed during measurements. Highly-oriented pyrolytic graphite (HOPG) was used for peak position calibration. Peaks were fitted with a Lorentzian function using Wire 4.4 software from Renishaw. Raman mapping was employed using high-speed encoded mapping stage to generate high definition 2D chemical images over typically around 140,000 μm$^2$. Step size was set at 3 μm to avoid over-exposure and damage.

C–H bonding was directly detected using Fourier transform infrared (FTIR) spectroscopy. FTIR images and spectra were recorded with an Agilent-Cary 670 FTIR spectrometer coupled to a Cary 620 FTIR microscope equipped with a 64 × 64 focal plane array mercury cadmium telluride detector. Attenuated total reflection (ATR) mode was employed, with a germanium crystal as internal reflection element, yielding a pixel resolution of 1.4 μm.

TEM (mostly electron diffraction) was employed to examine the material structure before and after hydrogenation. As opposed to the behavior of graphene or diamond, the converted material appeared to be quite electron sensitive, possibly because of the high degree of surface hydrogenation, even for electron energy as low as 80 keV [2]. This is consistent with the photon sensitivity observed under certain conditions [2] and the reported electron sensitivity of F-diamane [19]. For this reason, a very low voltage (5 kV) benchtop transmission electron microscope from Delong Instruments was used. The instrument includes a Schottky-type field emission gun and a 2560 × 2160 pixel Front Illuminated Scientific CMOS (6.5 μm$^2$ pixel size). Electron diffraction patterns were obtained in selected area mode from 100 nm-large areas.

### 2.4. Computational Details

The atomic structures, the quasi-particle band structures, and optical spectra were obtained from Density Functional Theory (DFT) calculations using the VASP package [23,24] and the plane-augmented wave scheme [25,26] to treat core electrons. Perdew–Burke–Ernzerhof (PBE) functional [27] was used as an approximation of the exchange-correlation electronic term for all the geometry optimization steps as well as for phonon and frequency calculations. The cut-off energy was set to 400 eV, with a Gaussian smearing with a width of 0.05 eV for partial occupancies. To build the wave-function, which served as starting point for further $G_0W_0$ calculations [28,29], the Heyd–Scuseria–Ernzerhof (HSE) exchange-correlation functional [30,31] was used. During the geometry optimization step, all atoms were allowed to relax with a force convergence criterion below 0.005 eV/Å using the van der Waals corrected scheme of Grimme et al. [32]. The optimized lattice parameters of graphane, bilayer-diamane, and three-layer diamanoid were 2.53, 2.52, and 2.52 Å, respectively. A vacuum height of 20 Å was used to avoid spurious interaction between periodic images of the different slabs. A 21 × 21 × 1 grid was set for $G_0W_0$ calculations, in conjunction with a total number of bands of 800, and an energy cutoff of 100 eV for the response function, after a check of the direct band-gap convergence, to be smaller than 0.1 eV in function of k-point sampling. To estimate phonon dispersion (7 × 7), supercells were used with a 3 × 3 × 1 or 7 × 7 × 1 grid for k-point sampling, in the Density Functional

Perturbation Theory (DFPT) framework, using the Phonopy code [33]. VESTA software was used to generate optimized structure pictures [34]. Satisfactory calculation of the phonon dispersion curve of a pristine graphene monolayer was used to validate the present computational setting [17]. The characteristic G band was located at 1572 cm$^{-1}$ [17]. C–H stretching mode frequency calculations were performed from a finite difference approach, using a displacement of 0.001 Å.

*2.5. Nomenclature*

Although diamane has been described as a full series of materials structurally derived from hydrogenated 2LG, for the purpose of clarity, we have proposed a new nomenclature [2]. The first structure that the hydrogenation of a graphene could form is actually graphane, which consists of a single-layer of a hexagonal network of sp$^3$–bonded carbon atoms in which each carbon is bonded to one hydrogen atom, alternately above and below the layer [35]. The next is genuine diamane, which should, strictly speaking, be limited to the two-layer structure wherein half of the carbons from the both layers are hydrogenated while the other half are bonded to each other, thereby covalently bonding the layers. Two structural configurations are possible, depending on whether the stacking sequence in the starting 2LG is AB or AA. The former will result in the diamond structure-based diamane (called diamane I in [1]), while the latter will result in the lonsdaleite structure-based diamane (called diamane II in [1]). As the number of layers increases, because the inner layers cannot be hydrogenated, diamane-related materials should be more accurately described as surface-hydrogenated diamond or lonsdaleite, or more generally diamanoids, in order to include any mixture of diamond polytypes. Whereas an AB-stacked 2LG can generate a diamond structure, an ABA-stacked 3LG cannot because the half of the carbon atoms of layer B which are available for bonding with the second layer A are located in front of a hexagon center of this layer, instead of another atom. On the other hand, the diamond structure can develop from the ABCA*etc.* stacking sequence, while lonsdaleite structure can develop from AAA*etc* stacking. As soon as a diamanoid is built from more than 2 graphenes, its structure can also result from a mixture of diamond and lonsdaleite. Because of this, with the exception of graphane and diamane (I and II), it is more relevant to designate the subsequent hydrogenated multilayer diamanoid structures with the number of layers equal to 3, 4, and so on up to "few", by the stacking sequence of the starting graphenes, as suggested in [3], for instance (ABB)D, (ABBA)D, (ABBC)D, (AABBCC)D, and so on (where "D" stands for "diamanoid"). Alternatively, as for graphene, the number of stacked layers could be stated without precise identification of the stacking sequence, i.e., 3LD, 4LD, 5LD, and so on up to FLD or MLD (where F/MLD stands for "few-/multi-layer diamanoid"). The limit of "few" cannot be given; calculations to identify the point at which MLD properties are indistinguishable from those of regular diamond or lonsdaleite have yet to be performed. We note that this nomenclature does not discriminate between hydrogenation of one surface or both. If necessary, 1H-FLD and 2H-FLD could be used to designate one-surface- and both-surface-hydrogenated few-layer diamanoid, respectively. In some conditions, hydrogenation might affect one surface only of the two-layer sp$^3$–C structure (AB or AA). Such material would no longer be genuine diamane but could be described as "surface hydrogenated bi-layer diamond or lonsdaleite", abbreviated as 1H-2D or 1H-2L, respectively. In the case of fluorinated material, "H" can be substituted by "F" in the proposed nomenclature; a similar substitution may be made for any other atom or functional group to which the surface carbon atoms could be bonded in lieu of H.

**3. Results and Discussion**

*3.1. Hot-Filament Process to Efficiently Hydrogenate Graphene Materials*

To achieve the above-presented route to convert 2LG into diamane, it is necessary to develop an efficient method to hydrogenate 2LG. Only few experimental studies and methods have been reported on this topic so far, as previously reviewed [2,36]. In any case, only partially hydrogenated material has been prepared; never fully hydrogenated [2,36].

With regard to gas phase methods, which are applicable for nanoelectronics and photonics applications, the best result is of about 10 at.% of hydrogenated carbon, a value estimated in Ref. [36] from the results published in Ref. [37]. It was hypothesized that the hot-filament process might constitute a very competitive method for the efficient hydrogenation of graphene sheets and the subsequent formation of diamane [2]. Hot-filament CVD has been employed for the industrial production of diamond films for about 40 years because it efficiently produces atomic hydrogen (H), which has been shown to play a critical role for the conventional synthesis of metastable diamond at low pressure from a dilute mixture of a hydrocarbon in $H_2$ [38–41]. In the hot-filament process, H is produced heterogeneously by the thermal decomposition of $H_2$ on the hot filament surface, and rapidly diffuses into the bulk gas. H recombination reactions are sufficiently slow at the typical process pressures (below 100 Torr) that most of the H diffuse to the reactor walls. H is present at super-equilibrium concentration throughout most of the reactor [41]. The effects of $H_2$ pressure and flow rate, filament temperature, and radial distance from the filament on the relative H concentrations, and the gas temperature profiles have been investigated in detail, in particular for the case of pure $H_2$ [41–47]. In the hot filament process, as compared to low pressure plasma techniques, the presence of ions accelerated toward the substrate, which can damage the graphene, is avoided. High kinetic energy ions in the plasma tend to etch the graphene film instead of participating in the hydrogenation process [48]. The hot-filament process was successfully used to grow crystalline nanodiamonds at low substrate temperature, below 300 °C, on temperature-sensitive substrates such as kapton®VN [49–51]. Notably, it was then used to conformally coat carbon nanotube bundles with diamond and SiC nanocrystals from solid carbon and silicon sources exposed to H at ~190 °C, before the nanotubes could be etched away [21,52]. Figure 1a displays a typical UV Raman spectrum of such a material (see the corresponding scanning electron microscopy image in Figure 1b), showing a sharp diamond peak at about 1325 cm$^{-1}$ from diamond nanocrystals at the surface of nanotube bundles. From the efficient production of H by the hot-filament process and the possibility to process graphene materials at low substrate temperature, it was hypothesized that it might be possible to efficiently hydrogenate 2LG using such a process, and to subsequently produce genuine diamane.

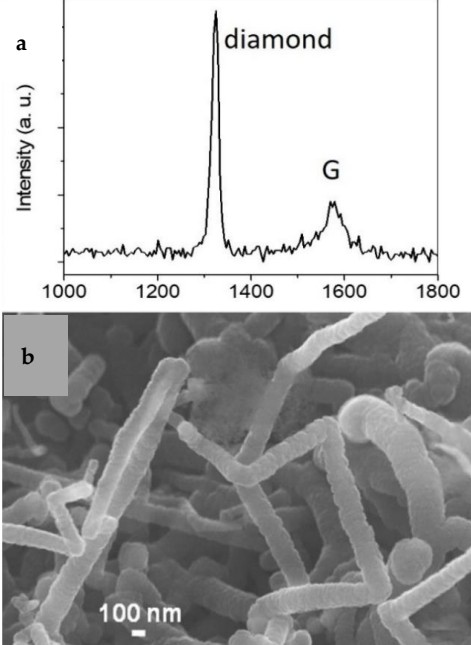

**Figure 1.** (**a**) Typical UV Raman spectrum of carbon nanotube bundles coated with diamond and SiC nanocrystals by the hot-filament process from solid C and Si sources; (**b**) corresponding scanning electron microscopy image. Modified from Piazza et al. Carbon 75 (2014) 113, Copyright Elsevier.

### 3.2. Genuine Diamane from 2LG

As evidenced by selected area electron diffraction, the pristine 2LG films present a lack of homogeneity regarding the stacking sequence and the number of domains covered by the aperture opening (Figure 2). Thanks to the use of an electron energy as low as 5 keV, which allows revealing pattern features different and somehow complementary to that obtained at the more common 80–100 keV energy [53], the following configurations are evidenced in the material:

(i)      single 2LG domain with AA stacking (illustrated by Figure 2a);

(ii)      single 2LG domain with AB stacking, which discriminates from 2LG-AA above by the three-fold symmetry of the spot intensity distribution on the first ring (illustrated by Figure 2b) [54];

(iii)      single 2LG domain made of two randomly stacked 1LG (illustrated by Figure 2c), otherwise designated as twisted 2LG;

(iv)      multiple 2LG domains, corresponding to the various combinations of cases (i) to (iii) above (illustrated by Figure 2d); this case happens when the selection aperture covers an area of the 2LG film where several neighboring domains separated by defect lines are present.

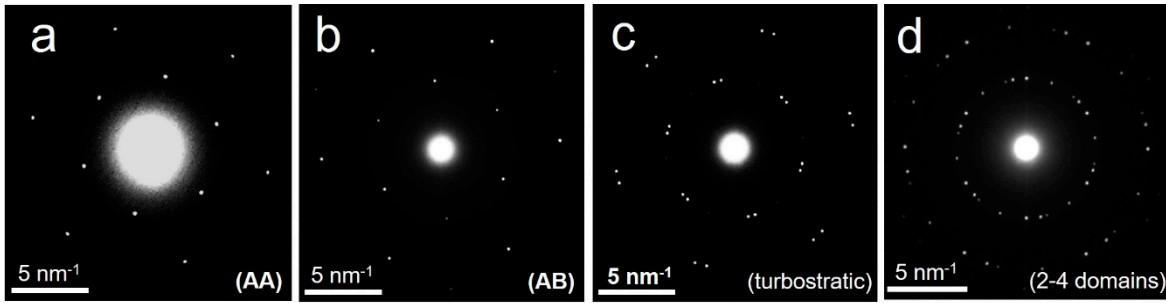

**Figure 2.** Examples of diffraction patterns obtained in various locations of the bi-layer graphene (2LG) films before hydrogenation. (**a**) Single graphenic domain, as an evidence for two superimposed single-layer graphene (1LG) in AA stacking; the inner and outer 6 spots correspond to the *100* and *110* spot families in graphite, respectively, corresponding to the 0.213 and 0.123 nm d-spacings. (**b**) The same, however with an AB stacking, as evidenced by the three-fold symmetry of the spot intensity distribution (see text). (**c**) The same, however with the two superimposed 1LG slightly rotated, as an example of a random (turbostratic) stacking; in principle, the pattern could also correspond to two neighboring 2LG-AA domains slightly misoriented, but considering how the 2LG films were prepared, the chance for getting the turbostratic stacking is much higher by far. (**d**) Multiple 2LG domains, as a combination of the various cases shown in (**a**) to (**c**); for this example which shows four 6-spot systems, explaining them by two neighboring 2LG domains with random stacking is likely; indeed, no three-fold symmetry of the spot intensity distribution is seen for any of the 6-spot systems, hence AB stacking is excluded; on the other hand, in principle, the pattern may also correspond to 4 misoriented 2LG-AA domains, or 2 misoriented 2LG-AA domains and 1 turbostratic 2LG domain; but as previously said, the chance for getting the AA stacking is much lower than random stacking. Modified from Piazza et al., Carbon 169 (2020) 129, Copyright Elsevier.

Micro-Raman (UV) mapping was performed before and after exposure to the hot-filament-promoted hydrogenation process to track any resulting structure conversion change. Before the hydrogenation process, the spectra of 2LG are characterized by a regular sharp G peak at around 1582 cm$^{-1}$, due to bond stretching of all pairs of sp$^2$–C in graphene sheets. After the hydrogenation process, drastic changes can be observed in the Raman spectra from various regions. Figure 3a presents a typical spectrum of such a region, which exhibits specific features:

(i)      the G peak is no longer detected;

(ii)      the D peak which originates from defects in graphene sheets [37] is still not observed;

(iii)      a sharp peak (full width at half maximum (FWHM) of around 10–33 cm$^{-1}$) at around 1344−1367 cm$^{-1}$ has appeared.

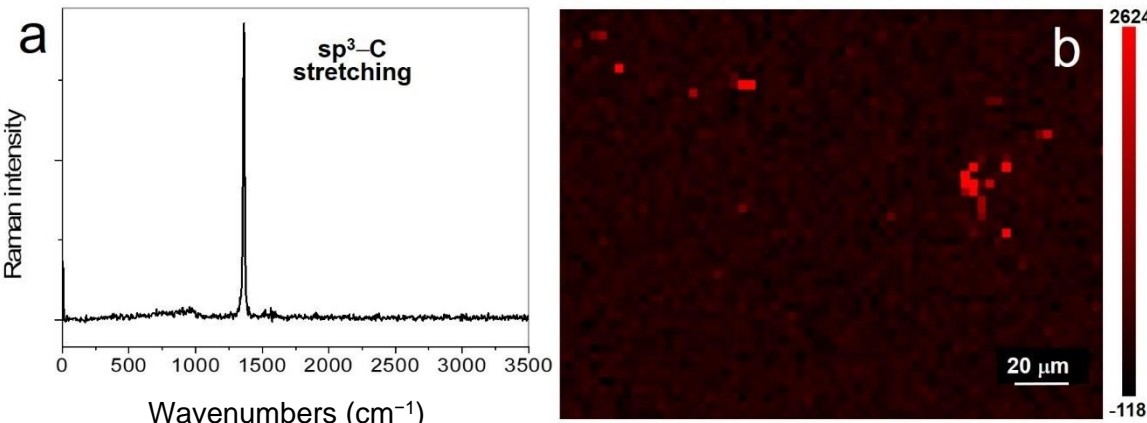

**Figure 3.** Typical UV Raman data for 2LG exposed to the hot-filament-promoted hydrogenation process. (**a**) Example of spectrum; (**b**) example of intensity map of the $sp^3$–C stretching mode. From Piazza et al., Carbon 169 (2020) 129, Copyright Elsevier.

This type of spectrum can be observed over extended regions of several μm width and in various areas probed by the laser (on both the grid gold-wire-supported material and the free-standing material), as illustrated in Figure 3b which displays a map of the intensity of this peak. The peak at around 1344–1367 cm$^{-1}$ is not the D peak, since the G peak is not simultaneously observed; moreover it is too sharp and its spectral position does not correspond to the value observed in the UV spectra of nanocrystalline graphite [54]. It is assigned to the diamond/lonsdaleite stretching mode between $sp^3$–C [2] (diamond $E_{2g}$ mode; lonsdaleite $A_{1g}$ and $E_{2g}$ modes). This mode was predicted to be near 1320 cm$^{-1}$ from ab initio calculations for diamane [4], and was found at slightly lower wavenumber in diamanoids (1319.5–1337 cm$^{-1}$) [2,17] and diamond (1332 cm$^{-1}$). Note that in lonsdaleite, the $A_{1g}$ and the $E_{2g}$ modes, corresponding to the triply-degenerated stretching mode of cubic diamond, are split into a component vibrating in the plane of the layers and a component vibrating perpendicular to the layers [55]. The bonding strength of lonsdaleite is comparable whether it is parallel or perpendicular to the layers and the wavenumber separation between the $A_{1g}$ and $E_{2g}$ modes is expected to be small [55]. We conclude that the results shown in Figure 3a,b are evidence that full $sp^2$–C to $sp^3$–C conversion, i.e., full conversion of 2LG into genuine diamane, can be obtained over a large region, here on the order of several tens μm$^2$.

It is assumed that the conversion takes place in regions of the 2LG film where graphene sheets are AB- or AA-stacked, as these are the most energetically favorable configurations [1,2]. As noted, the starting 2LG contains many domains with randomly stacked layers for which C–C interlayer bonding cannot occur; stress and/or strain is expected once these layers are hydrogenated, as well as at the grain boundaries between graphenic and diamane domains. This hypothesis is confirmed by the following observations:

(i)     Cracks have appeared in the film (compare Figure 4a,b).

(ii)    A range of up-shifted positions of the Raman $sp^3$−C peak are observed (Figure 5).

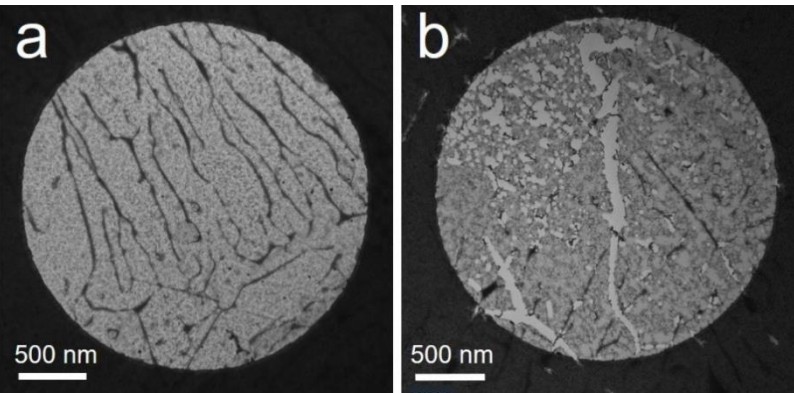

**Figure 4.** Examples of TEM image of 2LG films (**a**) before and (**b**) after hydrogenation. Contrast variations are due to the presence of amorphous material partially covering the film. Modified from Piazza et al., Carbon 169 (2020) 129, copyright Elsevier.

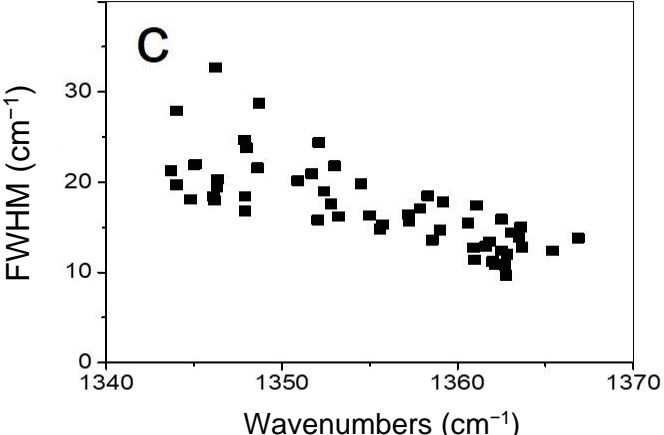

**Figure 5.** Full width at half maximum (FWHM) as a function of wavenumber of the Raman $sp^3$–C stretching peak for 2LG exposed to the hot-filament-promoted hydrogenation process. Modified from Piazza et al., Carbon 169 (2020) 129, Copyright Elsevier.

It must be pointed out that, under current knowledge, electron diffraction of such 2D films cannot be used as a sole proof of diamane structure, as the patterns do not allow discriminating with enough confidence between the graphene, diamond, and lonsdaleite structures, either from interplanar distances, spot intensity distribution, or from peak intensities, as shown in Ref. [17]. This limitation could however be overcome by using multiwavelength electron diffraction [53] but further calculations are still needed to apply the methodology to diamanes and diamanoids. However, in our case, the graphene to crystallized $sp^3$–C structure conversion is established from the Raman spectra.

In Figure 5, one can note a correlation of the Raman $sp^3$–C stretching peak width with its position, which ranges from 1340 to 1370 cm$^{-1}$. An upshift in FWHM could arise from the increased strain due to the presence of several interconnected phases (diamane, graphene), but could also be due to inhomogeneous layer hydrogenation, or to a distorted bonding between two one-side-hydrogenated twisted layers. Because a splitting between singlet and doublet due to hypothetical biaxial strain is not observed in the Raman spectra, the rough hypothesis of an average hydrostatic strain is assumed to be acceptable. As calculated in [56], the Grüneisen parameter for optical phonon in diamond and lonsdaleite is around 3 cm$^{-1}$/GPa. The in-plane lattice parameter is 0.249 nm in lonsdaleite, 0.252 nm in diamond, and 0.246 nm in graphene, leading to a strain of −1 to −2%. In graphene or diamond, −1% is associated to a hydrostatic stress of about 10 GPa.

Consequently, a wavenumber shift of 30 cm$^{-1}$ is possible and corresponds to the range of the shift observed.

The presence of stress or strain is further confirmed by first principle calculations on modified diamane structures. In the case of under-hydrogenation, achieved by removing 7 H atoms from the top layer of genuine diamane structure (one hydrogen atom every two carbon atoms), a strong structural reorganization occurs (Figure 6a,b). The 19 C atoms involved in the extended defect tend to lose their sp$^3$ character and to recover sp$^2$, implying the shortening of several C–C bond-lengths, confirmed by the histogram given in Figure 7.

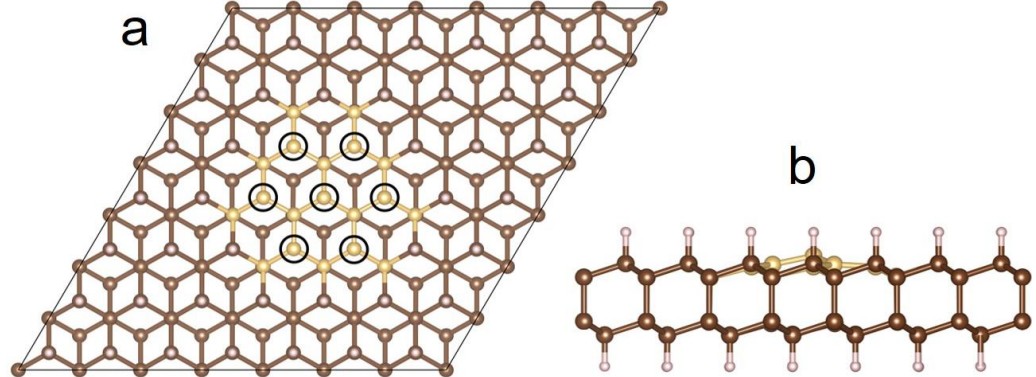

**Figure 6.** (**a**) Top view of the reconstructed, unsaturated structure of diamane in situation of under-hydrogenation, with seven missing hydrogen atoms on the top layer (the positions of the missing H atoms are given by the black circles), and (**b**) side view. Carbon atoms involved in the reconstruction are given in yellow, while the others are brown, and H atoms are white. Modified from Piazza et al., Carbon 169 (2020) 129, Copyright Elsevier.

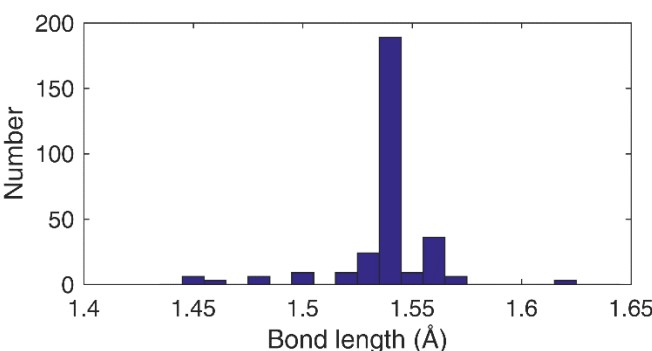

**Figure 7.** Resulting C–C bond-length distribution of the reconstructed unsaturated structure, with seven missing hydrogen atoms on the top, as shown in Figure 6. Modified from Piazza et al., Carbon 169 (2020) 129, Copyright Elsevier.

Thus, the lattice parameter remains the same while local distortions can be present which could be at the origin of both the linewidth broadening and frequency variations (Figure 5). These calculations show that significant distortions can occur upon partial hydrogenation. The sp$^2$–C to sp$^3$–C conversion induces huge stresses, specifically at the domain boundaries, either in the case of under-hydrogenation or when the 2LG involved does not exhibit the perfect AB or AA stacking needed for the inter-bonding of the layers. In the case of turbostratic or commensurate stacking, only partial sp$^2$–C to sp$^3$–C conversion may occur, locally promoted when carbon atoms happen to locally superimpose. This is also consistent with the stress estimated in diamanoid/graphene hybrids [17].

First principle calculations show that full hydrogenation for AB stacking is the most stable final material. First, starting from the genuine diamane structure, and then subsequently moving a hydrogen or a pair of hydrogen atoms to the neighboring C atoms,

a spontaneous relocation of the hydrogen atoms to the correct sites for diamane formation is observed. Thus, it appears that hydrogen atoms self-organize when reacting with a graphene layer. Second, in the case of C atoms already coordinated to 3 other C atoms, such that two first-neighbor C atoms both wear a hydrogen, addition of excess H atoms always results in spontaneous desorption to reform $H_2$ molecules. Over-hydrogenation of the structure is thus not energetically favorable. This is quite convenient for setting the synthesis parameters, as it indicates that an excess of hydrogen is preferable to an insufficient supply of it with no risk to affect the resulting diamane (or diamanoid) structure.

### 3.3. Diamanoid/Graphene Hybrid from FLG
### 3.3.1. Model of Diamanoid/Graphene Hybrid

To mimic our experimental diamanoid/graphene samples, which are made of a various number of layers above two, we modelled partially hydrogenated and converted systems consisting of four distinct carbon layers (Figure 8a) [17]. The first layer (L1) corresponds to the fully hydrogenated layer typical of diamane and diamanoid (all C atoms are sp$^3$–hybridized, half of them bonded with one H atom, the other half are bonded with C atoms from layer L2). The second layer (L2) is not hydrogenated, hence half of the C atoms are truly sp$^3$–C and bonded to C atoms from L1, while the other half are also sp$^3$-hybridized but exhibit a free orbital with one unpaired electron of $p_z$ character which interacts with layer L3 (Figure 8b). The two bottom layers L3 and L4 remain pure sp$^2$–C; however, L3 interacts more strongly with L2 than with L4 due to the existence of the unsatisfied valence bonds in the former. As a consequence, the L2–L3 interlayer distance is shorter (in the range 0.3037–0.3308 nm) than the L3–L4 interlayer distance (0.3512 nm). For thicker systems, other layers similar to L4 can be added underneath while respecting the stacking sequence in graphite (ABAB), until the other side of the flake is reached where the occurrence of the L1–L2–L3 combination may repeat. Despite the stacking sequence of graphite in our pristine FLG being highly probable, it is important to determine whether the sp$^3$–C to sp$^2$–C conversion generates stress in the system, sufficient to induce layer decoupling upon relaxation, more preferably between layers where the interaction is weaker (typically between L3 and L4). Therefore, different L1− ... −L4 stacking orders were tested, namely ABBB, ABAB, ABCA, and ABBA, the latter being more stable by at least 15 meV/carbon atom than the other stacking sequences. Only the ABBA stacking sequence is discussed below, but it is worth noting that the structure changes described below would be the same for ABAB stacking.

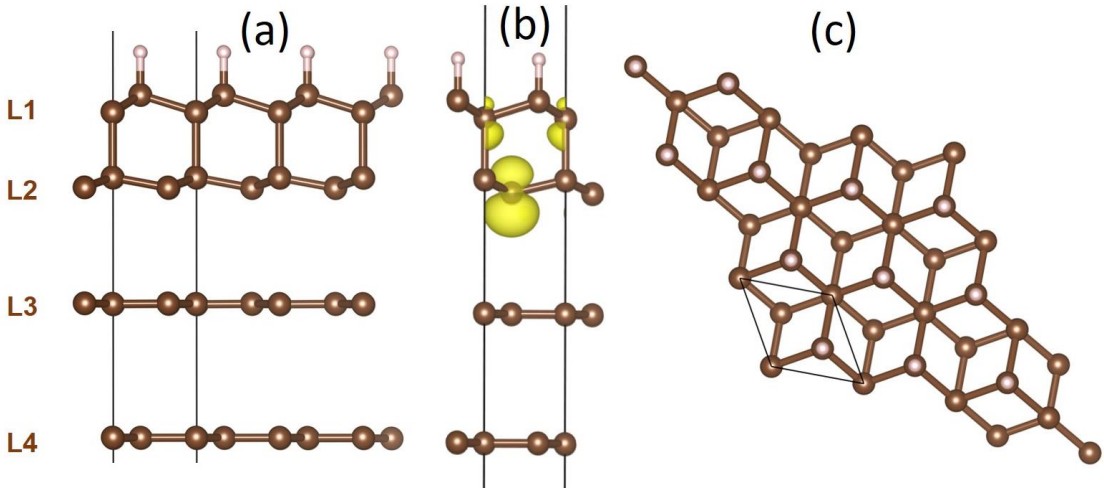

**Figure 8.** (**a**) and (**b**) two side-views and (**c**) top view of the partially hydrogenated few-layer graphene (FLG) used in Density Functional Theory (DFT) calculations with ABBA stacking. The projected structure is that of the face-centered cubic of diamond. In (**b**) is shown how the $p_z$ orbitals are preferably oriented towards the layer underneath (L3). The black lines delineate the primitive cell. From Piazza et al., Carbon 156 (2020) 234, Copyright Elsevier.

The model considered in Figure 8 is based on the idea that, when starting from FLG, the hydrogenation process generates the formation of an upper diamanoid domain involving L1 and L2 in the model (possibly added with L3, because of its stronger interaction with L2 than with L4). This domain lies upon a lower, untransformed FLG domain (involving L4 in the model and subsequent graphene layers underneath).

### 3.3.2. UV Raman Spectroscopy Analysis

As in the case of 2LG, before the hydrogenation process, the spectra of FLG are characterized by a regular sharp G peak at around $1582\ \mathrm{cm^{-1}}$. After the hydrogenation process, drastic changes can be observed throughout the Raman spectra. Figure 9a,d show representative corresponding UV Raman spectra. No difference was observed whether the spectra were obtained on the free-standing region or the copper-supported region of the FLG film lying on the TEM grid.

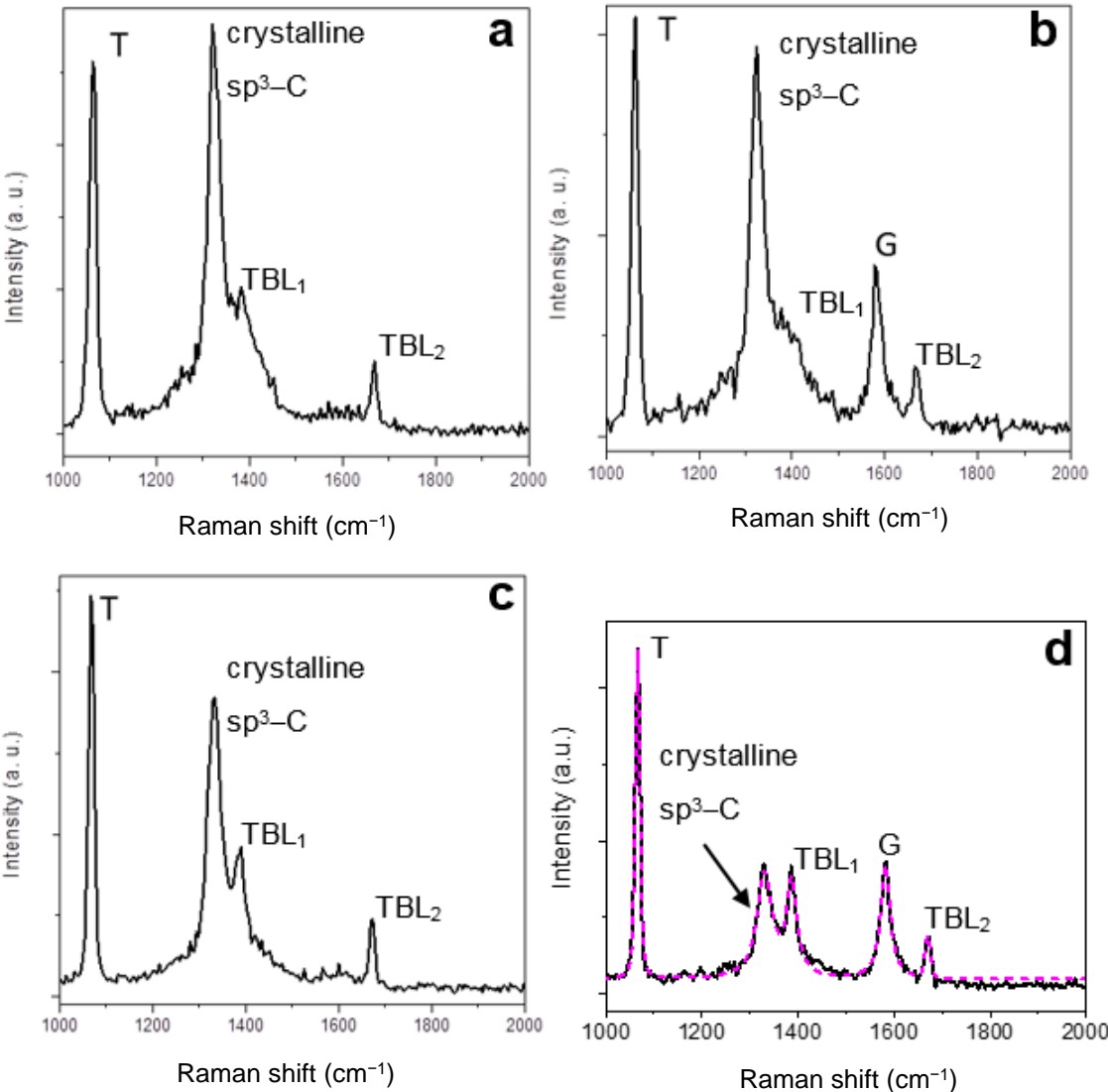

**Figure 9.** Typical UV Raman spectra of FLG after the exposure to the hot-filament-promoted hydrogenation process (**a**–**d**). The spectra show the crystalline $sp^3$–C peak at $1319$–$1337\ \mathrm{cm^{-1}}$. The G band reveals the partial conversion only (**b**) and (**d**). Two peaks typical of twisted 2LG (see text), labeled TBL$_1$ and TBL$_2$ (twisted bilayer), are also observed at $\sim$1385 cm$^{-1}$ and $\sim$1669 cm$^{-1}$. An example of the fitting made by using symmetric Voigt functions (dashed line) to estimate peak positions is provided in (**d**). Modified from Piazza et al., Carbon 145 (2019) 10, Copyright Elsevier.

The spectrum shown in Figure 9a displays two sharp peaks centered at ~1063.0 and ~1323.1 cm$^{-1}$, respectively. The first peak is interpreted as the T peak, due to C–C sp$^3$ vibration, previously found in tetrahedral amorphous carbon (ta-C) films as a low-intensity and broad band [57,58] or as a high-intensity and narrow peak in diamond nanoclusters [59]. The physical origin of the *T* peak will be further discussed below. Here, the T peak is narrow and of high relative intensity, indicating that the structure of the sp$^3$–C involved is crystalline, as opposed to the FWHM of the *T* peak in ta-C, which is wide because of the amorphous structure. As in the case of diamane, the peak centered at ~1323.1 cm$^{-1}$ is interpreted as corresponding to the diamond/lonsdaleite stretching mode (Section 3.2). In the pristine (i.e., before hydrogenation) FLG regions having more than two layers, hence multilayer domains, the Bernal stacking sequence ABA is likely, as the most frequent and the most stable (as compared to AAA and ABCA stacking). Hence, the diamond structure (face-centered cubic, FCC) should be favored. In the case of the cubic structure, the shift of the peak could be due to stress [38], or to confinement effects in few atomic layers or in sub-10 nm nanocrystals [60,61] or to the diamond/lonsdaleite hybrid structure. Those factors could also explain the relatively large peak FWHM value. This will be discussed further below. Notably, the T and crystalline sp$^3$–C peaks were observed in visible Raman spectra, yet attenuated [2]. Hence, the Raman spectroscopy results show that it is possible to form crystalline sp$^3$–carbon ultrathin sheets (i.e, diamanoid) from the hydrogenation of FLG at low temperature and low pressure and the subsequent formation of interlayer sp$^3$–C bonds.

Figure 10 displays examples of the intensity map of the sp$^3$–C stretching mode Raman peak. Figure 10a shows that the peak can be detected in multiples regions in a surface area of ~220 × 140 μm$^2$. Figure 10b,c show some examples of extended regions of ~33 × 51 μm$^2$ (Figure 10b), ~21 × 33 μm$^2$ (Figure 10c), and ~15 × 15 μm$^2$ (Figure 10c), where crystalline sp$^3$–C material is continuously detected. These dimensions are consistent with that of the hydrogenated region revealed by FTIR microscopy (Section 3.3.4). The laminar character of diamond/lonsdaleite will be further discussed below.

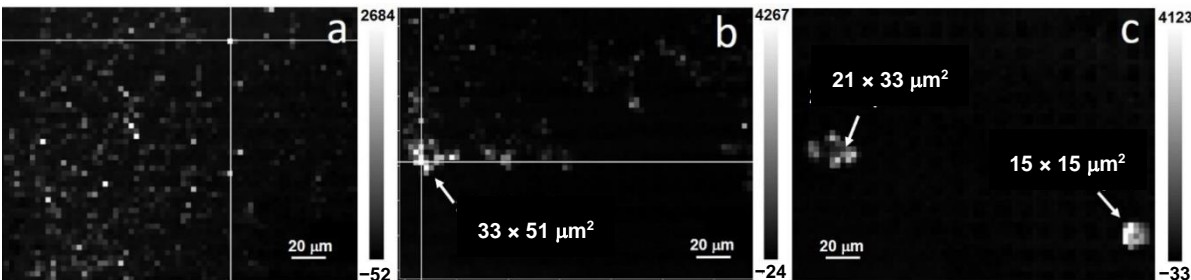

**Figure 10.** (**a–c**) Typical UV Raman maps showing the distribution and relative intensity of the diamond/lonsdaleite stretching mode Raman peak of FLG exposed to the hot-filament-promoted hydrogenation process. From Piazza et al., Carbon 145 (2019) 10, Copyright Elsevier.

No change was observed in the sp$^3$–C material over more than five years as checked by UV Raman spectroscopy. It is therefore assumed that converted crystalline sp$^3$–C layers, having Raman spectra such as that shown as Figure 9, are stable over time.

In most of the regions, the T and the diamond/lonsdaleite stretching peaks are simultaneously detected as shown in Figure 9. From the analysis of 329 spectra taken from four mappings and three samples, variations in both peak positions are observed. The positions of the T and crystalline sp$^3$–C peaks are found to vary within a range of 16.4 cm$^{-1}$ and 17.5 cm$^{-1}$, respectively and, more interestingly, both peak positions vary in parallel so that there is a clear correlation between both (Figure 11).

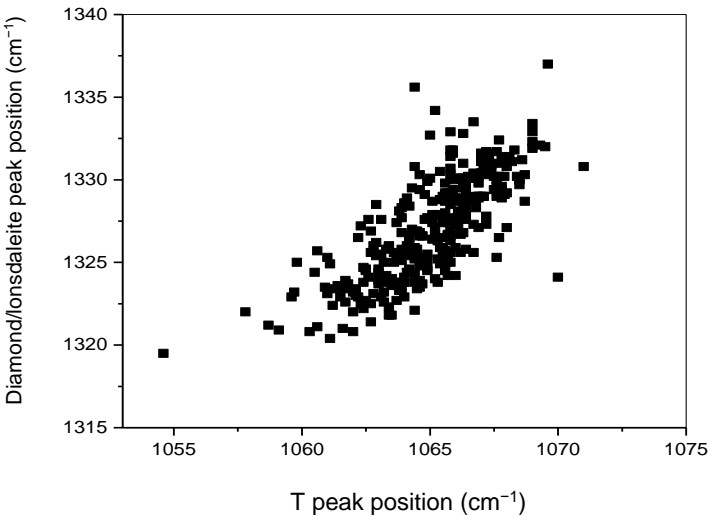

**Figure 11.** Crystalline sp$^3$–C mode Raman peak position as a function of the T peak position in FLG exposed to the hot-filament-promoted hydrogenation process (spectra taken at 244 nm). From Piazza et al., Carbon 145 (2019) 10, Copyright Elsevier.

The position of the crystalline sp$^3$–C peak is found to vary from 1319.5 to 1337 cm$^{-1}$, and thus includes 1332 cm$^{-1}$, the value for diamond [22]. However, this result does not allow a clear identification of the actual structure, as it could be consistent with the occurrence of both lonsdaleite and diamond as well as a variable combination of both. This result could also reveal the existence of stresses due to partial sp$^2$–C to sp$^3$–C conversion as in the case of hydrogenated 2LG, discussed above.

In some regions of the grid, the T and crystalline sp$^3$–C peaks are simultaneously detected along with the graphene G peak at ~1582 cm$^{-1}$ (Figure 9b,d). The relative intensity of the crystalline sp$^3$–C and G peaks varies across the sample probed. In some cases, the intensity of the *G* peak is several times higher than that of the *T* and crystalline sp$^3$–C peaks. Those observations suggest that it is possible to prepare graphene-crystalline sp$^3$–C hybrids from the partial conversion of FLG into crystalline sp$^3$–C material. The UV spectra (Figure 9) show two additional peaks labeled TBL1 and TBL2. The physical origin of those peaks will be discussed further below. Table 1 summarizes the frequency of the main peaks observed in UV Raman spectra of genuine diamane and diamanoid/graphene hybrid.

**Table 1.** Summary of the main peaks observed in UV Raman spectra of genuine diamane and diamanoid/graphene hybrids (see text).

| Material | sp$^3$–C Stretching (cm$^{-1}$) | T (cm$^{-1}$) | TBL$_1$ (cm$^{-1}$) | TBL$_2$ (cm$^{-1}$) |
|---|---|---|---|---|
| Diamane | 1344–1367 | N.A. | N.A. | N.A. |
| Diamanoid/graphene hybrid | 1319–1337 | 1055–1071 | 1385 | 1669 |

3.3.3. Electron Diffraction

As expected, as-received FLG films present a lack of homogeneity in the number of layers and stacking order (Figure 12). Figure 12a,d show examples of electron diffraction patterns taken from three different 100 nm-large areas. Figure 12a corresponds to a single domain (i.e., with the first and second rings bearing only 6 spots each), while Figure 12b–d are examples of areas comprising two, three and six coherent graphenic domains superimposed or possibly adjacent, respectively. Of over more than 53 grains analyzed, only 6 were found to exhibit the diffractogram of a single domain such as in Figure 12a.

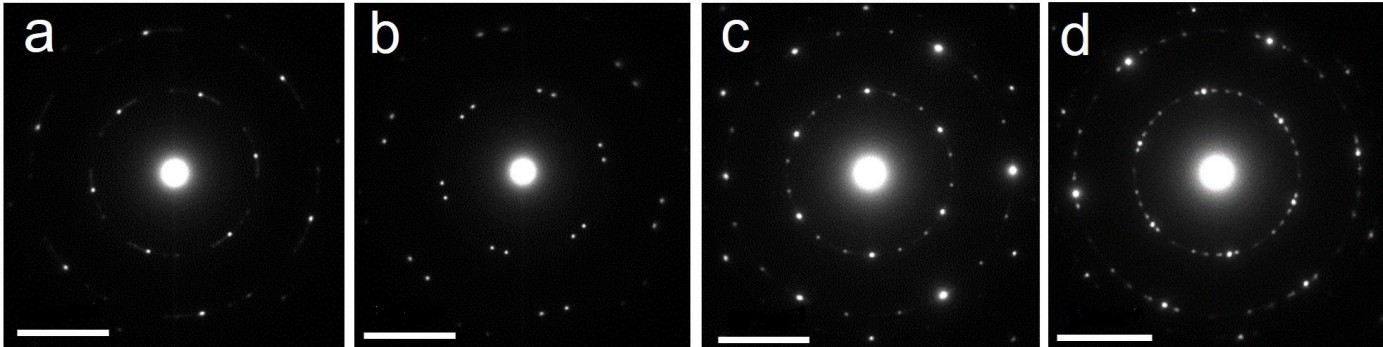

**Figure 12.** (**a–d**), typical electron diffraction patterns of pristine FLG, showing 1, 2, 3 and 6 adjacent or superimposed graphenic domains. Scale bar: 5 nm$^{-1}$.

After exposure to the hydrogenation process, electron diffraction patterns in some regions of the FLG differ from the typical patterns of pristine FLG and/or of pure ultrathin and crystalline sp$^3$–C sheets. Figure 13 displays examples of such patterns taken from different areas. With over 53 grains analyzed, such patterns were not observed in pristine FLG. Indeed, the patterns shown in Figure 13 are more complex, exhibiting additional peaks as satellites of the regular sets of 6 peaks per domain and per ring. Such satellite peaks frequently represent the periodicities of moirés, which occur when two coherent domains with similar periodicities superimpose with a slight rotation angle. The patterns from Figure 13 therefore reveal the occurrence of superimposed and slightly twisted coherent domains (TCD) with similar periodicities. This interpretation is fully supported by the observation of moiré patterns in TEM images (Figure 14).

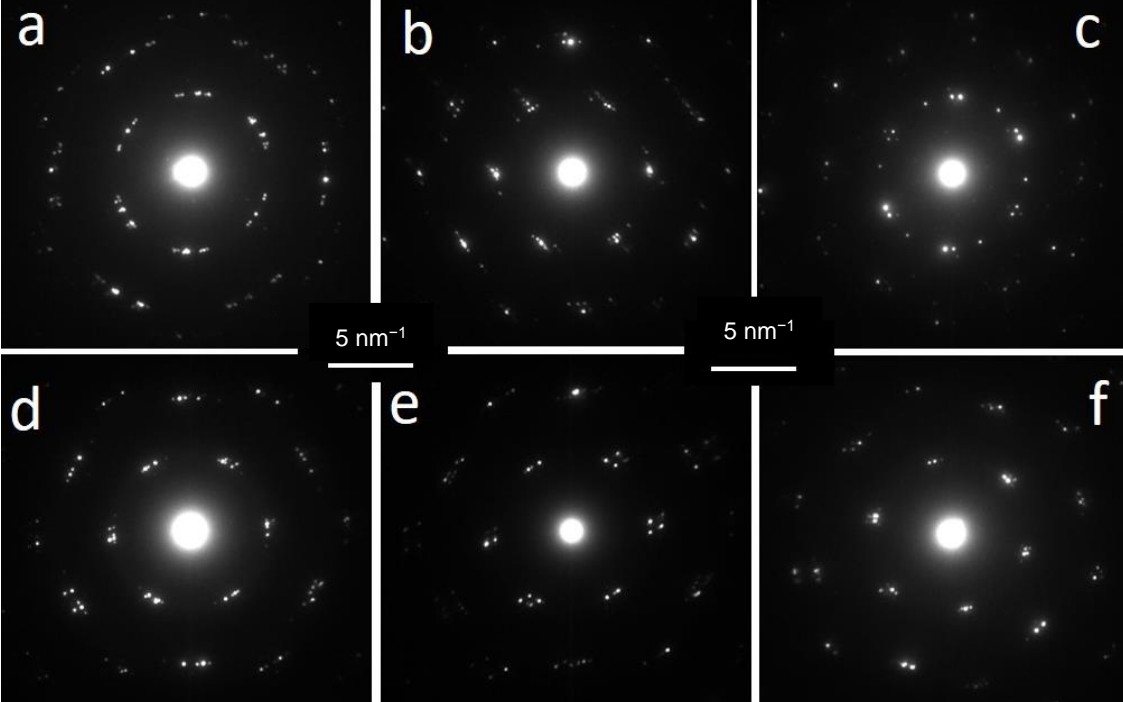

**Figure 13.** (**a–f**) Typical electron diffraction patterns of film areas after exposure to the hot-filament-promoted hydrogenation process. The satellite peaks which make the patterns more complex are characteristic of superimposed coherent domains with similar periodicities twisted with a small twist angle (designated as twisted coherent domains (TCD), see text). From Piazza et al., Carbon 156 (2020) 234, Copyright Elsevier.

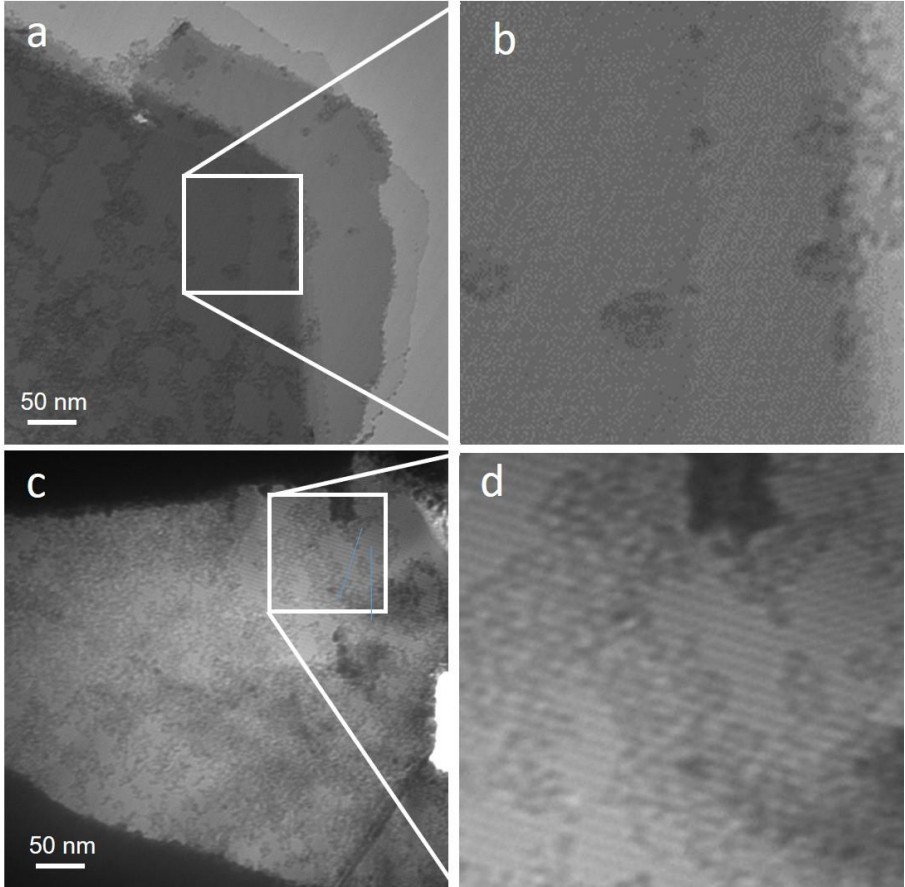

**Figure 14.** (**a**,**c**) are the TEM images of the specimen areas having provided the diffraction patterns shown in Figure 13e,f, respectively. (**b**,**d**) are magnified views of the framed areas in (**a**,**c**), to better reveal the moiré periodicities.

Referring to the model in Figure 8 (see Section 3.3.1), the TCDs found here could be the upper diamanoid domain (represented by L1 and L2, completed with L3) and the lower graphenic domain (represented by L4 and other graphenes underneath). From those results, we conclude that two graphene layers (L3 and L4) in twisted configuration are at the interface between the two twisted domains. The twisted bilayer (TBL) system has been recently studied [62,63]. This interpretation is supported by the detection of the specific peaks observed at ~1385 cm$^{-1}$ and in ~1669 cm$^{-1}$ in UV Raman spectra (labeled TBL$_1$ and TBL$_2$ in Figure 9 and Table 1), as reported for TBL [64].

It is supposed that the slight rotation of the superimposed domains from the TCDs is induced by the relaxation of huge (several GPa) local constraints that develop between superimposed layers as a result of the sp$^2$–C-to-sp$^3$–C conversion. In diamond-like carbon, a material that contains variable fractions of sp$^3$–C and sp$^2$–C, internal stress can reach several GPa [12,13,65]. In genuine diamane, compressive stress of the order of 30 GPa was inferred (see Section 3.2). This will be discussed further below.

### 3.3.4. FTIR Microscopy

To directly detect C–H bonding, FTIR mapping was carried out. Figure 15a shows a FTIR-ATR image processed on the integrated intensity of a sharp C–C stretching band at ~1608 cm$^{-1}$ that co-localizes with the lone C–H stretching mode observed at 2846 cm$^{-1}$. The image was obtained on the plain part of the TEM grid which circles the grid mesh, and which was exposed to H radicals. Prior to treatment, this region of the grid was found to contain FLG as evidenced by Raman spectroscopy analysis. After hydrogenation, a large-size circular area, of ~150 µm$^2$, containing C–H bonding is revealed (Figure 15a).

This first confirms that C–H bonding was generated during the hydrogenation process and, second, indicates that hydrogenation took place in the basal plane of graphene, not only at the edges of graphene domains. The dimension of the area containing C–H bonds is consistent with those of the regions where the $sp^3$–C stretching mode Raman peak is detected (Figures 3b and 10b,c).

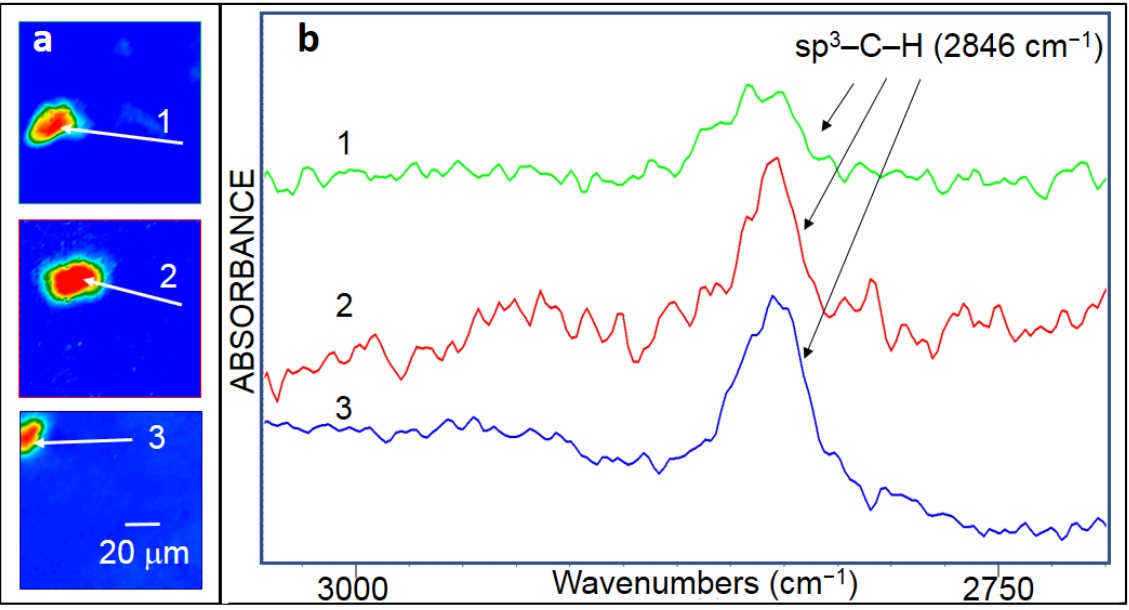

**Figure 15.** (**a**) Typical FTIR-ATR microscopy images processed on the integrated intensity of the associated C–C stretching band at ~1609 $cm^{-1}$ that co-localizes with the C–H stretching band of FLG exposed to the hot-filament-promoted hydrogenation process. The false colour image scale runs from high (red) to absent (blue). (**b**) Typical absorbance FTIR spectra (sums of 10 to 25 spectra per image) taken from pixels within the red regions arrowed in (**a**). The number of each spectrum in (**b**) refers to numbered arrow in (**a**).

Figure 15b displays one absorbance spectrum (cumulated over 5 spectra) taken in the region identified by a white arrow in Figure 15a. It shows that the narrow C–H stretching band, centered at around 2846 $cm^{-1}$, is composed of only one among the nine possible vibration mode components. It is proposed that this mode corresponds to the $sp^3$–C–H stretching mode. This assignment is based on several arguments, which were detailed in ref. [2]. Briefly:

1.  The symmetrical $sp^3$–C–$H_2$ and symmetrical $sp^3$–C–$H_3$ modes are excluded as they would be simultaneously detected with their antisymmetrical counterparts.
2.  The olefinic $sp^2$–C–H and aromatic $sp^2$–C–H modes are excluded as they are typically detected at much higher wavenumber, above 2975 $cm^{-1}$ in the case of free molecules [66].
3.  The symmetrical and anti-symmetrical olefinic $C_2H_2$ modes are excluded as they are typically detected at significantly higher wavenumber, above 2950 $cm^{-1}$, in the case of free molecules [66]. Furthermore, they would be accompanied by their anti/symmetrical counterpart, which is not the case here.
4.  Our experimental value is consistent with the calculated C–H stretching mode frequency for graphane, 2H-(AB)D, (AB)D, and 2H-(ABC)D [2]. The C–H stretching mode frequencies of isolated H on graphene and on single vacancy in graphene are found to be much higher giving a further evidence of high hydrogenation rate.

The detection of the $sp^3$–C–H stretching mode is a remarkable result since the IR absorption cross-section for this mode is expected to be significantly lower than the values for the other modes, as in free molecules [66]. The shift, by around 54 $cm^{-1}$, of the position of

the sp$^3$–C–H mode peak, as compared to the corresponding position in free molecules [66], is theorized to result from the structure change generated by the hydrogenation of FLG, the subsequent conversion of sp$^2$ hybridization into sp$^3$ hybridization, and the interlayer sp$^3$–C bond formation as further discussed below.

The narrow, one-component C–H stretching band reveals that carbon atoms are bonded to one hydrogen atom, corresponding to a single configuration, which is expected for diamane and diamanoid surface. This single component narrow C–H stretching band has never been reported before in the case of hydrogenated graphene. Known disclosures on FTIR spectroscopy analysis of hydrogenated graphene report on a wide multi-component C–H stretching band including sp$^3$–C–H$_3$ and sp$^3$–C–H$_2$ modes instead of sp$^3$–C–H mode [67–73], consistent with graphene domains of reduced size which are hydrogenated only on their edges. Comparatively, the relatively intense one-component narrow C–H stretching band shown here is representative of large-size (~150 μm$^2$) graphene hydrogenated on planes so that the related FTIR signal prevails over that generated by the hydrogenated graphene edges. Should the sp$^3$–C phase be under the form of crystalline clusters, the sp$^3$–C–H$_3$ and sp$^3$–C–H$_2$ vibration modes would have been detected in FTIR spectra, especially as their IR absorption cross-sections are expected to be significantly higher than for the sp$^3$–C–H mode, as in free molecules [66]. This strongly suggests that at least part of the 2D character of the material has been maintained upon the sp$^2$ to sp$^3$ conversion.

### 3.3.5. Understanding Further the Characterization Results—Modeling

We further investigated the physical origin of the sharp T peak, which is, in most cases, simultaneously detected with the sharp diamond/lonsdaleite stretching mode peak in UV Raman spectra and which was assigned to bonding between sp$^3$–C (Figure 9). First, we verified that the T peak is due to sp$^3$–C. For this purpose, a region of 2LG exposed to the hot-filament promoted hydrogenation process where the T peak was the only one detected in Raman spectra, was over-exposed to the UV laser in order to induce a possible structure transformation [2]. In fact, the measurement was repeated on the same spot under the same conditions [2]. After the second measurement, the G peak appeared in the spectrum, together with the T peak [2]. This experiment confirms that the T peak is from sp$^3$–C.

No T peak is present for pure sp$^2$–C and sp$^3$–C materials. There is a spatial correlation between the occurrence of the diamond/lonsdaleite stretching mode peak and that of the T peak (Figures 9 and 11) while there is no spatial correlation between the location of the G peak and that of the T peak. Thus, we have hypothesised that the T peak could be related to the interface between a sp$^2$–C layer and a sp$^3$–C layer, and consequently we have investigated the phonon dispersion of the ABBA-stacking model proposed in Figure 8 (see Section 3.3.1). Upon full atomic relaxation, all frequencies remain positive, proving the stability of the proposed structure.

In addition to two modes accounting for the crystalline sp$^3$–C peak in the Raman spectra (blue triangle in Figure 16), three distinct Raman active modes in the range of interest are present, with wavenumbers from 1050 to 1100 cm$^{-1}$ (green square in Figure 16). Normal mode analysis shows that two of them, corresponding to a large rotation of hydrogen atoms (Figure 17), are degenerated with a wavenumber of 1099 cm$^{-1}$ at the center of the Brillouin zone (Γ point) and are visible in Raman spectra. Interestingly, the mode at 1078 cm$^{-1}$, which involves atoms from L1 and L2, can be associated to a combination of the stretching of the sp$^3$–C bonds with an optical out-of-plane (ZO) mode of graphene membrane.

Due to the periodic repetition of the motif in DFT calculations, delamination is not possible in the model; thus, the full cell relaxation leads to compressive strain in the sp$^3$–C part (L1–L2) and to extensive strain in graphene layers underneath (L3–L4, and further layers underneath, if any), explaining the position of the G band (red dot in Figure 16) which is downshifted by almost 150 cm$^{-1}$ in the calculation compared to the pristine graphene. Using the conversion factor of 4.5 cm$^{-1}$/GPa [74], the stress brought to the

graphene domain (L4 and further layers underneath) can be estimated at $-33$ GPa. Notably, this is the same range inferred in the case of 2LG exposed to the hydrogenation process (Section 3.2). Obviously, this is a huge stress value, which is by far enough to generate the delamination and twisting events that are experimentally observed in the material, although the periodic conditions in the calculation cannot account for them. The system exhibits a perfect, periodic interface contrary to amorphous materials, consequently, the experimental linewidth of the T band is very narrow, at 14.0 cm$^{-1}$.

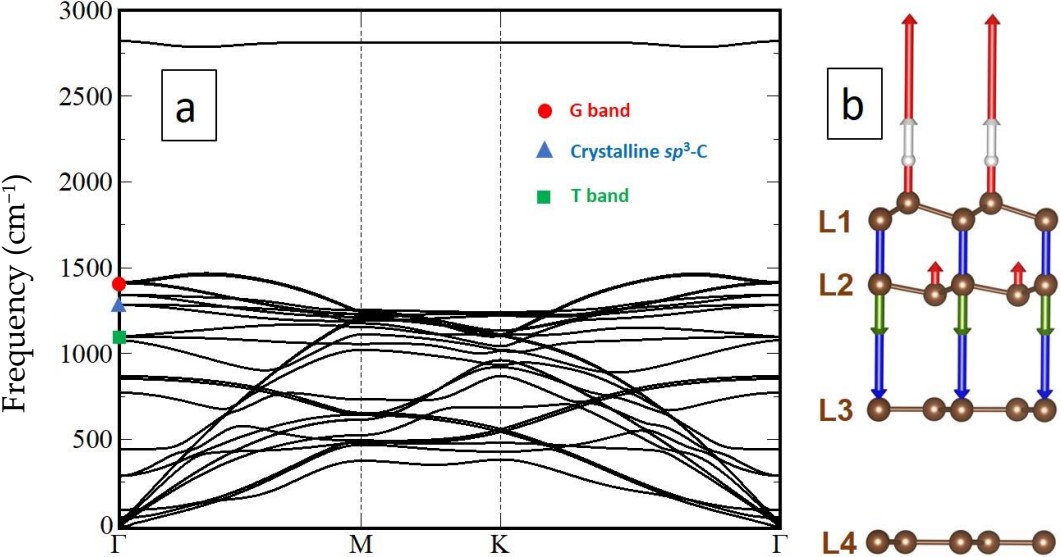

**Figure 16.** (**a**) Phonon dispersion curve of the partially hydrogenated ABBA stacked FLG model proposed in Figure 8, based on Density Functional Perturbation Theory (DFPT) calculations. (**b**) Corresponding vectors associated to the atomic displacements along the normal mode at 1078 cm$^{-1}$. Vector lengths are not at the scale of the model but are proportional to each other. Modified from Piazza et al., Carbon 156 (2020) 234, Copyright Elsevier.

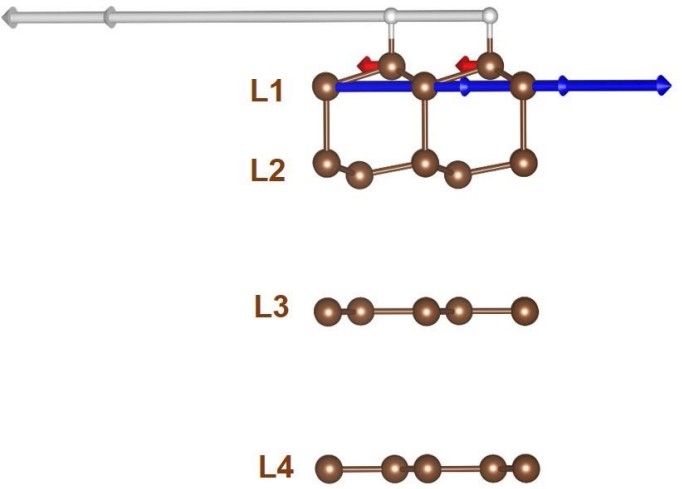

**Figure 17.** Vectors associated to the atomic displacements along one of the two degenerated normal modes at 1099 cm$^{-1}$; the other is simply anti-symmetric in the $x$-$y$ exchange. This figure is complementary to Figure 16. The two modes at 1099 cm$^{-1}$ correspond to a large rotation of H atoms. Since C atoms from L2 are not involved, these modes cannot account for the experimental spatial correlation between the T peak and the sp$^3$–C stretching peak (Figures 9 and 11), contrary to the mode reported in Figure 16. All modes are Raman active.

In summary, the model presented in Figure 8 along with our DFT calculations are consistent with the experimental characterization results. In particular, the model is able to account for the five peaks observed in Raman spectra in the 1000–1800 cm$^{-1}$ wavenumber range. The results show that it is not possible to form pure diamanoids starting from FLG with ABA*etc* stacking as soon as the number of layers is higher than two. In the latter case, only the first two layers will be sp$^3$–C, of which only the surface layer is in fully-coordinated configuration (on both sides of the FLG). Furthermore, only the first surface layer will be sp$^3$–C (on both sides of the FLG) if the starting FLG is turbostatic. The formation of diamanoids with more than two layers requires starting from FLG with AAA*etc* or ABC*etc* stacking; such direct synthesis is unlikely because both are energetically unfavorable. The only way could be to enforce such stacking sequences by piling up individual single crystal graphenes with a full and accurate control of the mutual orientation angle during the deposition. On the other hand, the synthesis of diamane is possible starting with 2LG for any but the turbostatic stacking sequence.

### 3.4. An Additional Route for the Formation of Diamond Grains in Space?

Clarification is still needed for the origin and formation mechanism(s) of meteoritic and interstellar diamond dust grains, including those supposed to be presolar [75]. As reviewed in Ref [76], "opinions regarding the formation mechanism(s) of meteoritic and interstellar diamond grains have changed over the years as new methods of making synthetic diamond were developed . . . ". Our results reveal the possibility of an additional formation route wherein graphene grains are converted into diamane or diamanoid by exposure to H flux without the requirement for high pressure. Those conditions can actually exist in space, for instance in the circumstellar envelope of C-rich evolved stars. Such a formation route of diamond-related structures in space is even more possible since the necessary pre-existence of graphene layers in space is very likely, as suggested by:

(i) the (possible) extragalactic detection of planar C$_{24}$ from the *Spitzer* Space Telescope [77]; recent DFT and coupled-cluster calculations on wavefunction stability showed that the graphene form of the C$_{24}$ was the most stable of different types of C$_{24}$ isomers, including the fullerene form, and that it best accounts for the astronomical data [78]; and

(ii) the recent speculation from laboratory experiments that FLG grains could be formed in space from pyrene [79].

## 4. Conclusions

We have shown that the hot-filament-promoted hydrogenation process can be successfully used to produce genuine diamane from bi-layer graphene at low pressure and at low temperature. It is believed that the key for producing homogenous diamane films is to start with very high quality 2LG material, ideally, single-crystal AB flakes as large as possible. However, the multi-domain 2LG we started with contains a large proportion of randomly stacked layers, for which C–C interlayer bonding cannot occur, and a lower proportion of AB and AA domains for which the sp$^2$–C to sp$^3$–C conversion may happen. We demonstrated the existence of large stress (tenths of GPa) once these layers are hydrogenated, and at the grain boundaries between graphenic and diamane domains. We have shown that it is possible to produce diamanoid/twisted bi-layer-graphene hybrids by using the same process while replacing 2LG by FLG. In both cases (diamane and diamanoid), UV Raman spectra exhibit the sp$^3$–C stretching mode peak of crystalline sp$^3$–C material. When the starting material happens to be FLG instead of 2LG, a twisted bi-layer graphene configuration is evidenced by characteristic Raman peaks ("TBL" peaks), and by complex electron diffraction patterns including satellite peaks. They are presumably formed following the relaxation of the stress resulting from the partial sp$^2$–C to sp$^3$–C conversion, estimated at around 33 GPa by DFT calculations. The frequent evidence of the only partial conversion of sp$^3$–C to sp$^2$–C confirms that conditions for producing diamanoids are not met when the starting material is FLG with more than two layers with the stacking sequence other than AAA*etc.* or ABC*etc.*, which are unfortunately the thermodynamically unfavored

configurations. However, the Bernal sequence (ABA*etc.*) may be used for the fabrication of graphene/diamanoid heterostructures, for which electronic properties have not yet been explored. The relation between electronic properties and the twist angle between the two first $sp^2$–C layers underneath the upper two diamane-related layers is of particular interest.

Significantly, our DFT calculations revealed the exact origin of the T peak, which has been reported in the literature but not explained. It was simultaneously detected with the $sp^3$–C stretching mode peak in the case of diamanoid/graphene hybrids and originates from a combination of the $sp^3$–C stretching mode of a $sp^3$–C layer with the optical out-of-plane mode of a graphene layer. Both layers are actually sandwiched between a highly hydrogenated $sp^3$–C surface layer and the underlying unconverted graphene layer (s); their strong interaction is due to the presence of free $p_z$ orbitals in the $sp^3$–C layer.

The successful conversion of a graphene layer into a hydrogenated $sp^3$–C layer comes with the occurrence of a single C–H stretching band in hydrogenated FLG, ascribed to the prevalent presence of $sp^3$–C–H groups over any other group involving more than one H. In our converted material, such a mode was detected by FTIR over sample surfaces as large as 150 $\mu m^2$.

Raman mapping and FTIR microscopy observations indicated the hydrogenation and subsequent $sp^3$ conversion over surface areas of up to 2000 $\mu m^2$; further studies are needed to ascertain $sp^3$–C domain dimensions. Importantly, it is believed that dimensions are only limited by the dimensions of the starting material, not by the process. Therefore, the present results open the door to mass production of diamanes, diamanoids, and diamanoid/graphene hybrids (including twisted bilayer-graphene configurations) for a wide range of applications [80], by means of the hot-filament-assisted CVD process, the well-established method for the industrial production of other $sp^3$–bonded carbon materials, such as diamond films. Finally, this work may provide new insight into the origin of some crystalline $sp^3$–bonded carbon grains in extraterrestrial environments.

## 5. Patents

The following patent applications were filed: U.K. Patent Application Number 1809206.4, June 5, 2018; Taiwan Patent Application Number 108117148, May 17, 2019; and P.C.T. Patent Application Number PCT/EP2019/064233, May 31, 2019.

**Author Contributions:** Conceptualization, F.P., M.M., P.P., and I.C.G.; methodology, F.P., M.M., P.P., I.C.G., and K.G.; software, P.P. and I.C.G.; validation, F.P., M.M., P.P., I.C.G., and K.G.; formal analysis, F.P., M.M., P.P., I.C.G., and K.G.; investigation, F.P., M.M., P.P., and I.C.G.; resources, F.P., M.M., P.P., and I.C.G.; data curation, F.P., M.M., P.P., I.C.G., and K.G.; writing—original draft preparation, F.P.; writing—review and editing, F.P., M.M., P.P., I.C.G., and K.G.; visualization, F.P., M.M., P.P., I.C.G., and K.G.; supervision, F.P.; project administration, F.P. and M.M.; funding acquisition, F.P., M.M., and I.C.G. All authors have read and agreed to the published version of the manuscript.

**Funding:** This research was funded by the Ministry of Higher Education, Science and Technology of the Dominican Republic, MESCyT (2010–2011, 2012, 2015, 2016, 2018 FONDOCyT programs). It has also been partially supported through the EUR grant NanoX n° ANR-17-EURE-0009 in the framework of the "Programme des Investissements d'Avenir" and the French Embassy in the Dominican Republic for mutual scientific visits and research stays. Allocations of computer time was funded by the Calcul en Midi-Pyrénées initiative CALMIP (Project p0812), as well as GENCI-CINES and GENCI-IDRIS for Grant No. 2019-A006096649.

**Institutional Review Board Statement:** Not applicable.

**Informed Consent Statement:** Not applicable.

**Acknowledgments:** F.P. would like to greatly acknowledge R. Bormett from Renishaw for technical support with Raman spectroscope; C. Ozoria, G. Paredes, and K. Cruz for technical support (Raman spectroscopy); and PUCMM for strong administrative support. R. Wiens is acknowledged for technical assistance in the FTIR microscopy experiments.

**Conflicts of Interest:** The authors declare no conflict of interest. The funders had no role in the design of the study; in the collection, analyses, or interpretation of data; in the writing of the manuscript, or in the decision to publish the results.

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
