# Peer review of "Progress on Diamane and Diamanoid Thin Film Pressureless Synthesis"

_carbon, 2020_

Round 1
Reviewer 1 Report
The review "Progress on Diamane and Diamanoid Thin Film Pressureless 2
Synthesis" is well written and it is very interesting for the topic.
I suggest to the editor to accept the manuscript in the present form.
Author Response
We thank Reviewer 1 very much for his/her time and his/her positive comments.
Reviewer 2 Report
Authors reported a very detailed and outstanfing work on syntheis and characterization by both experimental and DFT claculation o of diamane.
The work is not so innnovative fo synthetic approach but it is quite remarkable for the characterizationa and DFT study.
Nontheless, i stri gly suggest to fitt the spectra reported in Figure 9 or by using Voigt or by using lineshape reported by Tagliagerro et al. (Tagliaferro, A., Rovere, M., Padovano, E., Bartoli, M., & Giorcelli, M. (2020). Introducing the Novel Mixed Gaussian-Lorentzian Lineshape in the Analysis of the Raman Signal of Biochar. Nanomaterials, 10(9), 1748.). Furhtermore, do not use any acronyms in the abstract. Rewritten it accordingly.
I also encouraged the authors to add a section for a briefly comparison between the methodology that they reported and other previsously reported in literature. By using teh good that collected in this paper, a critical comaprison could be added to improve the quality of the paper.
COnsidering the points raised above, this paper deserves to be published by major revisions are required. I warly recommand to the authors to improve the paper because i think that this work could represent a referece point for the field.
Author Response
Reviewer #2:
Authors reported a very detailed and outstanfing work on syntheis and characterization by both experimental and DFT claculation o of diamane.
The work is not so innnovative fo synthetic approach but it is quite remarkable for the characterizationa and DFT study.
Our reply:
We thank Reviewer 2 very much for his/her time and his/her quite positive comments.
*******************
Rev #2:
Nontheless, i stri gly suggest to fitt the spectra reported in Figure 9 or by using Voigt or by using lineshape reported by Tagliagerro et al. (Tagliaferro, A., Rovere, M., Padovano, E., Bartoli, M., & Giorcelli, M. (2020). Introducing the Novel Mixed Gaussian-Lorentzian Lineshape in the Analysis of the Raman Signal of Biochar. Nanomaterials, 10(9), 1748.).
Our reply:
We thank Reviewer 2 for this suggestion. We agree with Reviewer 2 that a detailed fitting analysis would be of interest. However, we understand that such analysis is not necessary in our review manuscript as the features are clearly identified, for instance by fitting with symmetric Voigt functions. This simple analysis enables the determination of peak position. In order to take into account Reviewer 2 suggestion, we modified Figure 9 in the revised manuscript to include an example of fitting with symmetric Voigt functions (Figure 9d).
*******************
Rev #2:
Furhtermore, do not use any acronyms in the abstract. Rewritten it accordingly.
Our reply:
We have corrected this issue in the revised manuscript.
*******************
Rev #2:
I also encouraged the authors to add a section for a briefly comparison between the methodology that they reported and other previsously reported in literature. By using teh good that collected in this paper, a critical comaprison could be added to improve the quality of the paper.
Our reply:
Assuming that the "methodology" mentioned by the reviewer refers to the characterization procedure, we agree with Reviewer 2 that characterization techniques are critical. A critical review on this point was included in the introduction of reference 2 of the manuscript. It is not included here because we are focusing in reviewing our work. However, on the one hand, we pinpoint some important drawbacks of the techniques used for other diamane-related works such as in reference 19 (line 83-85). On the other hand, we rationalize the use of UV excitation to perform Raman spectroscopy (lines 145-153) and the use of low-electron energy to perform electron diffraction (lines 175-180 and lines 294-296). We understand that the advantages of our characterization methodology appear clearly in our manuscript, as noticed by Reviewer 2
*******************
Rev #2:
COnsidering the points raised above, this paper deserves to be published by major revisions are required. I warly recommand to the authors to improve the paper because i think that this work could represent a referece point for the field.
Our reply:
We thank Reviewer 2 very much for his/her positive comments.
Round 2
Reviewer 2 Report
The paper match the requirements for pubblication